

# Hysteresis between groundwater and surface water levels indicates the states of hydrological turnover affecting solute transport and redox processes

Lars Bäthke[1] and Tobias Schuetz[1]

[1] Department of Hydrology, Faculty of Regional and Environmental Sciences, University of Trier, Trier, Germany
*Correspondence to:* Lars Bäthke (baethke@uni-trier.de) and Tobias Schuetz (tobias.schuetz@uni-trier.de)

**Abstract**

Small streams are highly sensitive to variations in discharge, a sensitivity predicted to increase in future climate scenarios,

impacting ecological health of streams and water management practices. Prolonged low-flow conditions alter groundwater-surface water (GW-SW) exchange patterns, leading to extended losing phases and a reduced duration of gaining periods. This study examines the relationship between hydrological turnover (HT) and hysteresis patterns under various system states in a third-order tributary of the River Mosel in Trier, Germany, using high-resolution hydrological and chemical data collected over two years.

Our results reveal distinct seasonal dynamics in GW-SW exchange. Counterclockwise hysteresis, prevalent during summer and drought conditions, was linked to the expansion of the hyporheic zone and bank storage, which reorganizes flow paths and influences redox dynamics. We established a strong correlation between HT and hysteresis characteristics, identifying the h-index as a valuable diagnostic tool for tracking seasonal changes in GW-SW connectivity, storage and hyporheic zone behavior based on hydraulic preconditions.

As climate change intensifies drought conditions, the hyporheic zone will play a vital role in solute cycling and GW-SW connectivity. The h-index, combined with chemical and hydrological monitoring, provides a robust framework for understanding and predicting these dynamics in small stream ecosystems.





## 1 Introduction

The interaction between GW and SW systems has an impact on the dynamics of streamflow, solute transport, and nutrient cycling, with significant implications for water resource management and ecological health. Within the broader hydrological system, delays in streamflow response are closely linked to stream-catchment connectivity governing the timing and magnitude of water, solute, carbon, and nutrient exchanges between surface and groundwater. In combination, these factors significantly impact stream water quality and a catchment's response to rainfall events (e.g., Brunner et al., 2009, 2011; Covino et al., 2011; Zuecco et al., 2019). Connectivity patterns frequently manifest hysteretic behavior, a non-linear, loop-like relationship between dependent and independent variables such as groundwater levels and river state (e.g., Pavlin et al., 2021; Camporese et al., 2014; Penna et al., 2011; Outram et al., 2014; McGuire & McDonnell, 2010; Gelmini et al., 2022; Zuecco et al., 2016, 2019).

Hysteresis describes how a system's response depends on both current inputs and prior states and are defined by non-linear behavior in response to inputs (e.g. Phillips 2003; Camporese et al., 2014). This "hysteretic" behavior, commonly observed in interactions between groundwater and surface water levels, reveals the complex interplay among storage capacities, groundwater hydraulics and flow dynamics (Zuecco et al., 2019). For example, hysteresis can manifest as a delayed groundwater response to surface water discharge variations (Zuecco et al., 2016; Pavlin et al., 2021), highlighting the impact of the variability of hydraulic gradients between steam and aquifer within a catchment (Welch et al. 2015). Temporally asynchronous water level changes characterize subsurface flow paths and generating characteristic hysteresis loops between surface and groundwater (Penna et al., 2011). An improved understanding of hysteretic behavior thus might enhance our understanding of stream flow exchange with the riverbanks and the hyporheic zone, a nonlinear mechanism involved in stream flow generation and its chemical signature (McGuire & McDonnell, 2010).

Within the context of GW-SW interaction, hyporheic exchange fluxes are modulated by event characteristics and groundwater preconditions (Trauth & Fleckenstein, 2017). Defined as the boundary between ground and surface waters, the hyporheic zone expands and contracts depending on the relative difference between groundwater levels and stream discharge (e.g., Wroblicky et al., 1998; Arntzen et al., 2006). Thus, significantly affecting exchange volumes (Candenas & Wilson, 2007; Malzone et al., 2016). An increase in stream stage typically increases the hyporheic volume, whereas it shrinks during recession or low flow conditions (Soulsby et al., 2001). Hyporheic exchange in the riparian zone induced by fluctuations in the stream stage are referred to as bank storage. This extends the hyporheic volume further into the stream bank, distinct from in-stream hyporheic exchange (Cooper & Rorabaugh, 1963). Bank storage often induces transport and mixing of chemical species between groundwater and surface waters, leading to processing of these solutes and a delayed return of these potentially modified chemicals (Gu et al., 2012). Bank storage-induced GW-SW mixing thus has the potential to alter solute fluxes out of the hyporheic zone, influencing the chemical signature of groundwater discharge (McCallum et al., 2010).



The interaction between stream and groundwater levels propagates surface water fluctuations into groundwater systems, enhancing GW-SW exchange (Xin et al., 2018). Tools such as the hysteresis index developed by Zuecco et al. (2016) and similar methods by Lloyd et al. (2016b) systematically quantify and compare hysteresis during runoff events. These tools support the classification and characterization of hydrological responses in catchments (Gelmini et al., 2022). However, stream flow responsiveness depends on multiple factors, such as local groundwater levels, with stream water level fluctuations modifying hydraulic gradients that drive GW-SW interactions at the stream-groundwater interface (Boano et al., 2014; Cardenas, 2008).

Several studies advocate that the exchange between GW and SW along streams needs to be addressed as bidirectional, consisting of gross gains and losses, with the cumulative bidirectional flux understood as a hydrological turnover (HT). HT is a major control on solute transport, influences in-stream ecological functions. Thus, modifying stream chemical signatures, making it a critical framework for understanding the bidirectional GW-SW interactions explored in this study (Payn et al., 2009, 2012; Covino et al., 2011; Mallard et al., 2014; Jimenez-Fernandez et al., 2022; Jähkel et al., 2022; Bäthke & Schuetz, 2024).

The process of GW-SW interaction in headwater streams integrates the movement of water masses between the near-stream aquifer, the hyporheic zone, and the stream channel (Payn et al., 2009; Ward et al., 2013, 2019). The link between bidirectional GW-SW interaction processes and local small scale groundwater gradient change, expressed in the hysteresis between near-stream groundwater and stream water levels, has not yet been fully explored.

Addressing this gap, this study examines how seasonal variations of stream stage and groundwater levels, hydraulic gradients, and bank storage extension, in conjunction with hysteretic behavior, influence hydrological turnover (HT) in a headwater stream. By quantifying hysteresis loops and HT over two seasons, we determine the impact of groundwater-surface water exchange variability on solute transport and stream chemistry. Through 68 hysteresis loops and 28 HT measurements, including nine during the 2020 drought, we analyze how HT and hysteresis characteristics respond to seasonality and hydraulic gradients. Stream and near-stream groundwater sampling (DOC, $NO_3^-$, Mn, Fe) reveal changes in redox zonation with hysteretic behavior. Silica serves as a proxy for groundwater, providing insights into subsurface contributions to stream chemistry. Potassium, as a local proxy for stream water, highlights the influence of GW-SW interactions on mixing and solute transport through the stream bank.



## 2 Materials and Methods

### 2.1 Study Area

The Olewigerbach is a third-order tributary of the River Mosel, located south of the city of Trier in southwest Germany. The catchment drains a 25 km² watershed with a total stream length of 14 km (Krein & Schorer, 2000). The elevation difference between the headwaters and the mouth is approximately 300 m. The stream has a pluvial regime, receiving an average annual precipitation of 745 mm and exhibiting a mean discharge of 106 l/s recorded between 2010 and 2023 at the study site. The catchment's geology is dominated by Devonian schists, primarily argillaceous slates. The shallow soil layer (~1.3 m) contains impermeable clay pockets underlain by colluvial or bedrock argillaceous slates (Krein & Symader, 2000). The study area is situated in the headwaters of the catchment at an elevation of approximately 290 m above sea level, characterized by steep hillslopes.

Drought conditions were defined as periods when streamflow fell below 5% (5.3 l/s) of the mean discharge, following thresholds set by Yevjevich (1967) and Zelenhasic & Salvai (1987). Using this definition, three severe drought periods were identified: March 27 to April 1, 2020; April 24 to September 28, 2020; and September 6 to October 20, 2021 (Figure 3).

Precipitation and modelled soil moisture data (AMBAV, DWD) for the study area were obtained from the Deutscher Wetterdienst (DWD) weather station located near the Olewigerbach catchment (Station: Trier-Petrisberg; ID: 05100; https://opendata.dwd.de/climate_environment/). Streamflow data were monitored at the sampling site (Figure 1).

**Figure 1: Spatial representation of topographic slopes of the Olewiger Bach catchment in the south of the city of Trier, Germany. Red: Gauging station with detailed sampling site sketch, with stream and sampling well setup. HT concept and groundwater connectivity through riparian zone. Blue: Location of the weather station. Grey: Location of additional sampling wells downstream.**

### 2.2 Experimental Setup

Field measurements were conducted from 2020 to 2022. The dataset comprises 29 differential discharge gauging campaigns, utilizing NaCl slug injections, to estimate hydrological turnover (HT) during runoff events, over the stream section equipped with sampling wells. In total, 68 hysteresis loops were recorded based on stream water levels and groundwater levels measured at two groundwater wells (GW1 and GW2) located close to the stream (Figure 1). Groundwater levels were continuously logged in 10-minute intervals using Orpheus Mini Level Loggers (OTT GmbH).

At the upstream gauging station three piezometers were installed in a transect: one in the stream (289.47 m a.s.l.) and two at the groundwater wells, GW1 (288.63 m a.s.l.) and GW2 (288.65 m a.s.l.). GW1 is located 1.5 m from the streambank, and



GW2 is 3.7 m away. Groundwater levels were monitored at a depth of 1.3 m. From the hydrograph, 68 events were selected for hysteresis analysis using the h-index developed by Zuecco et al. (2016). Corresponding HT measurements were available for 29 of these events. We collected 144 water samples—48 each from the stream, GW1, and GW2—across three identical

sampling spots, including two additional downstream locations without piezometers. The experimental infrastructure is drawn out in detail in Bäthke & Schuetz (2024).

**Tracer Injection Protocol:**

Instantaneous NaCl tracer injections were performed for HT estimation using dilution gauging (Day, 1976; Covino et al.,

2011; Mallard et al., 2014; Bäthke & Schuetz, 2024). Electrical conductivity during tracer injections ranged from 203–320 µS/cm at baseline, with peaks 100–300 µS/cm above baseline levels.

- Injection mass ranged from 50 g to 2000 g of pre-dissolved NaCl, adjusted based on discharge and background conductivity.

- Breakthrough curves (BTCs) with peaks outside 1.25–2.0 times baseline conductivity was excluded or repeated.

- Measurements covered discharge rates from ~1 to 400 l/s.

- Conductivity was logged at 1-second intervals, except during extreme low-flow conditions (<5 l/s), when a 5-second resolution was used to conserve device storage capacity.

Electrical conductivity data were used to calculate mass equivalents, which were calibrated to determine HT (Eq. 1). The hyporheic zone covers the space between the riparian groundwater and the stream (Wondzell et al., 2011). Bäthke & Schuetz

(2024) show that high HT magnitudes manifest in low variability between stream and groundwater silica concentrations, with silica concentrations serving as a proxy of prolonged contact with the subsurface (see as well e.g. Burns et al., 2003). Prior measurements have shown elevated potassium concentrations in the stream compared to groundwater, with its source upstream of the sampling site. Soils containing sufficient amounts of clay minerals (e.g. illite, vermiculite) may reduce the potassium in solution via sorption (Sparks & Huang, 1985). Thus, we measured potassium as a proxy for stream water. To

study redox changes within the riparian zone, we conducted additional sampling for DOC, $NO_3^-$, Mn, Fe. Sampling was shortly before tracer experiments at the wells and the stream (n=48). Lab measurements were conducted after filtration through a 0.45 µm glass fiber filter (Machery & Nagel). Nitrate was measured by ion chromatography (± 2%), potassium (± 5%), silica (± 8%), iron and manganese (± 5%) measurements were carried out by atomic absorption spectroscopy (AAS, contrAA 300), dissolve organic carbon (DOC) by combustion followed by IR detection (TOC analyzer, ± 2%).


### 2.2.1 Hydrologic Turnover

The conducted tracer experiments provided BTCs to quantify net changes in discharge along the reach (~250 m) as well as gross gains and losses of stream water to and from groundwater. With discharge estimation at the upper and lower end of the selected stream reach:





$$Q_i = \frac{M_i}{\int_0^t C_{M_i}(t)dt} \tag{1}$$


Where $Q_i$ [l/s] is the discharge at location $i$, $M_i$ [g] is initial tracer mass injected and $C_{M_i}(t)$ [g/l] the integrated tracer concentration from the BTC. Hydrological turnover (HT) assumes that the fractional loss of tracer mass represents the fractional loss of stream flow to the subsurface and does not enter the stream channel again during the duration of the experiment (Payn et al., 2009; Covino et al., 2011). Gross losses ($Q_{Loss}$) were calculated by:

$$Q_{Loss} = Q_{i-1} \frac{M_i - \int_0^t M(t)dt}{M_i} \tag{2}$$

Net discharge changes ($\Delta Q$) and HT were then determined as:

$$\Delta Q = Q_{Gain} + Q_{Loss} \tag{3}$$

$$HT_a = |Q_{Loss}| + Q_{Gain} \tag{4}$$

Finally, HT is normalized to a relative value based on reach length (d) in m:

$$HT = \frac{\frac{HT_a}{Q_{i-1}} \, 100}{d} \tag{5}$$

**2.2.2 The Hysteresis Index**

The hysteresis index (h-index) quantifies differences in water levels between rising and falling limbs of a hysteresis loop and is calculated in four steps (Zuecco et al., 2016). The extraction of stream water level data was done manually. Computation is done by utilizing the python script established by Jehn (2019, https://github.com/zutn/Hysteresis-Index-Zuecco). The script contains all computational steps as follows:

**Normalization:**

$$u(t) = \frac{x(t) - x_{min}}{x_{max} - x_{min}} \tag{6}$$



$$v(t) = \frac{y(t) - y_{min}}{y_{max} - y_{min}} \qquad (7)$$

**Integral computation** for rising ($A_{r[i,j]}$) and falling ($A_{f[i,j]}$) limbs, with A as area of the normalized Integral:

$$A_{r[i,j]} = \int_i^j v_r(u)\,du \qquad (8)$$

$$A_{f[i,j]} = \int_i^j v_f(u)\,du \qquad (9)$$

**Difference computation**:

$$\Delta A_{[i,j]} = A_{r[i,j]} - A_{f[i,j]} \qquad (10)$$

**Summation of differences** to obtain the h-index:

$$h = \sum_{k=1}^n \Delta A_{[i,j]} \qquad (12)$$

The h-Index value is quantified by the sum of the area within the normalized loop $\Delta A_{[i,j]}$, where n is the number of chosen intervals. With x as the GW-level and y as the SW-level at time t. Positive h values indicate clockwise hysteresis, while negative values indicate anticlockwise hysteretic behavior. A value equal to zero indicates no hysteresis or a symmetrical eight shaped. The extent of the hysteretic loop is given by the absolute value of h; the larger the hysteretic loop, the further h is away from zero (Zuecco et al., 2016; Gelmini et al., 2022).

### 2.2.3 Local Hydraulic Gradient

We assessed the local near stream hydraulic gradients in three different ways. First, we calculated the hydraulic gradient at each time step during the selected hysteresis events. The mean gradient from start to end of the hysteresis event is then associated to the corresponding hysteresis event, resulting in one mean hydraulic gradient per well during each hysteresis event $i_{GW_{(1,2)}}$[-] with, $h_S$ [m] as the mean piezometer height in the stream, $h_{GW(1,2)}$ [m] as the mean piezometer height at the wells GW1 and GW2 and the distance between the stream and the respective well as $d_{(1,2)}$[m].

$$i_{GW_{(1,2)}} = \frac{1}{n} \sum_{k=1}^n \frac{h_S - h_{GW(1,2)}}{d_{(1,2)}} \qquad (12)$$

Second, we calculated the mean hydraulic gradient $i_{GW_{(1,2)}}$ [-] between both groundwater wells during the defined hysteresis events:

$$i_{GW} = \frac{1}{n} \sum_{k=1}^n \frac{h_{GW1} - h_{GW2}}{2.2} \qquad (13)$$



## 3 Results

The majority of the events analyzed in summer showed groundwater levels of the near stream well (GW1) below the stream water level (Figure 2). However, in winter groundwater is often observed to be higher than the stream, especially at events

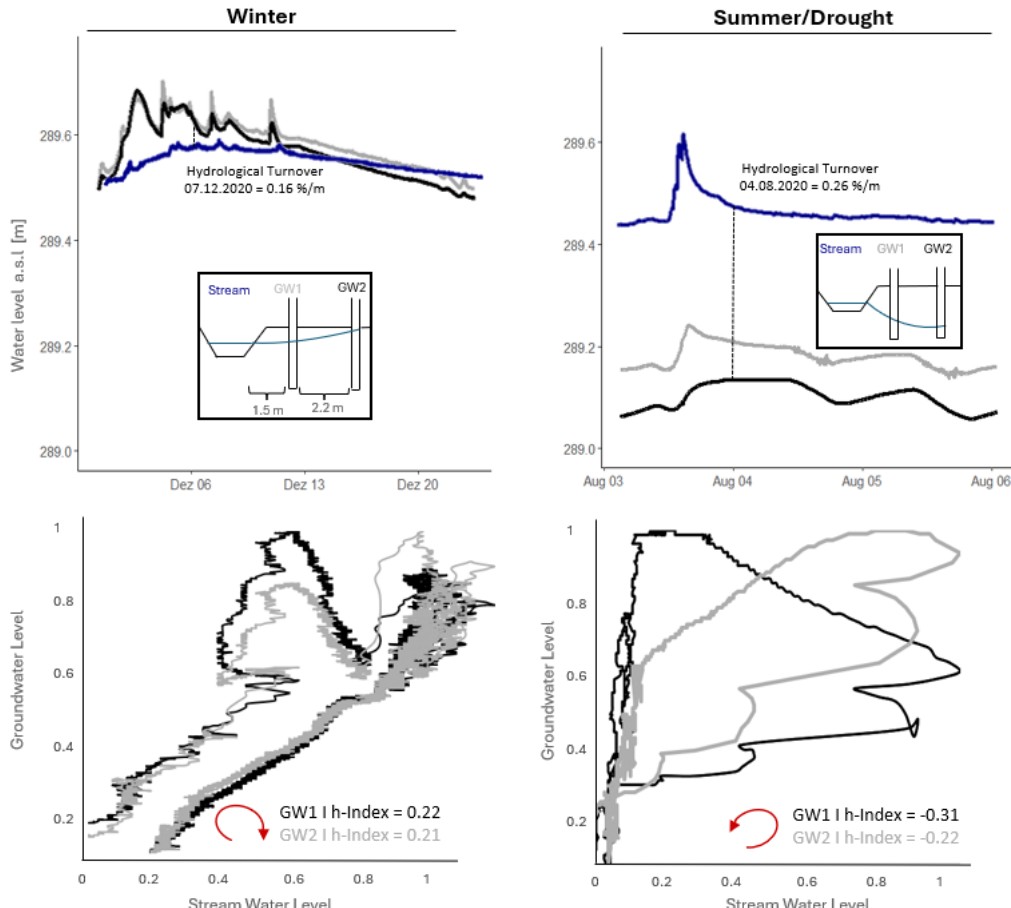

**Figure 2: Upper panels: Exemplary events of recorded water-table hysteresis between stream and wells. Left: Characteristic Winter water levels (blue: stream, black: GW1, grey: GW2). Sketch of the experimental setup. Time of HT measurement marked by dashed line. Lower panels: Normalized water levels plotted against each other as hysteretic loops. *h*-Index of the presented hysteretic loops of GW1 (black) and GW2 (grey). Right: Characteristic Summer water levels (blue: stream, black: GW1, grey: GW2). HT measurement marked by dashed line. Normalized water level plotted against each other as hysteric loop. h-indices of the presented hysteretic loops of GW1 (black) and GW2 (grey).**





during high soil moisture conditions. Analyzing the hysteresis between the stream and the respective groundwater well, we observed differences between hysteretic behavior during all seasons. During summer events the hydraulic head becomes negative, but also the hydraulic head difference between the two observed wells (Figure 2).

Precipitation events correspond with the on-site logged hydrograph dynamics as well as the soil moisture data (Figure 3 c).

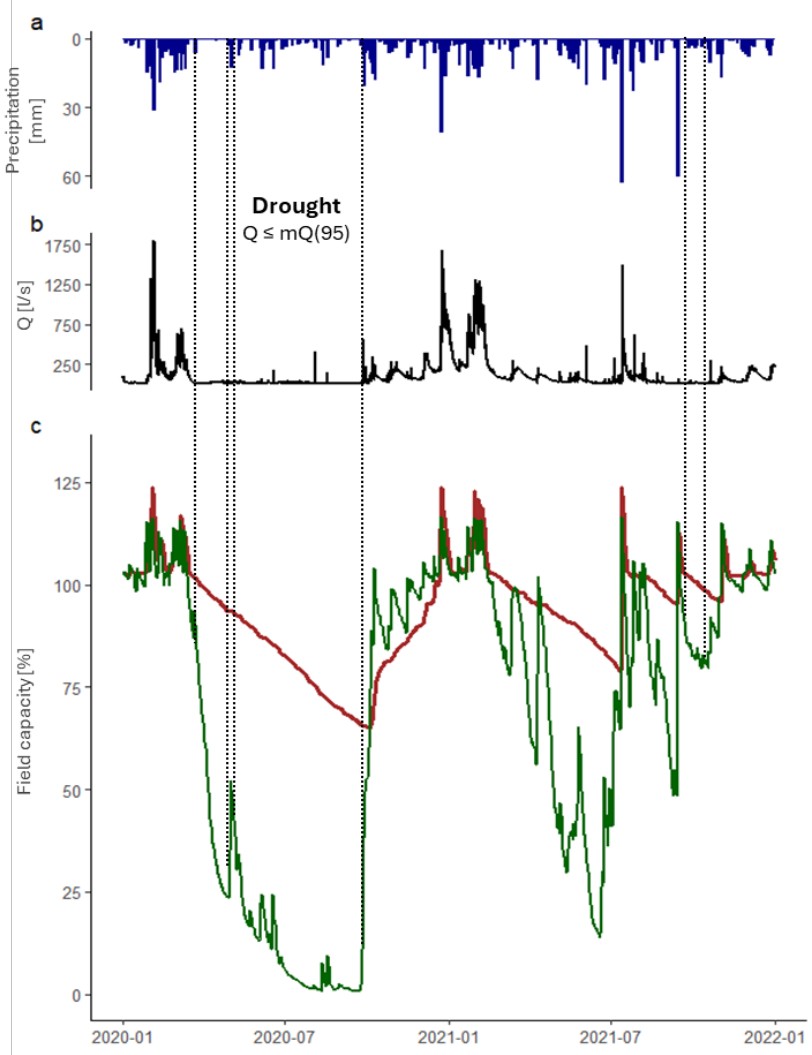

**Figure 3: (a) Catchment precipitation (mm) derived from a near weather station. (b) On- site stream flow (l/s). (c) soil moisture from a near weather station, two depths: 0-30 cm (green) and 30-60 cm (brown). Data presented over the observation period, with black dotted lines marking the identified periods of drought (Q below 95% of mean Q).**





Higher catchment soil moisture contents corresponded to higher stream flow and hydrograph reaction velocity to rain events.

During drought conditions at stream flow below 5% of mean discharge, we observed the highest HT [%/m] values (Figure 3 & 4), while absolute net exchange presents itself as very low (Figure 4).

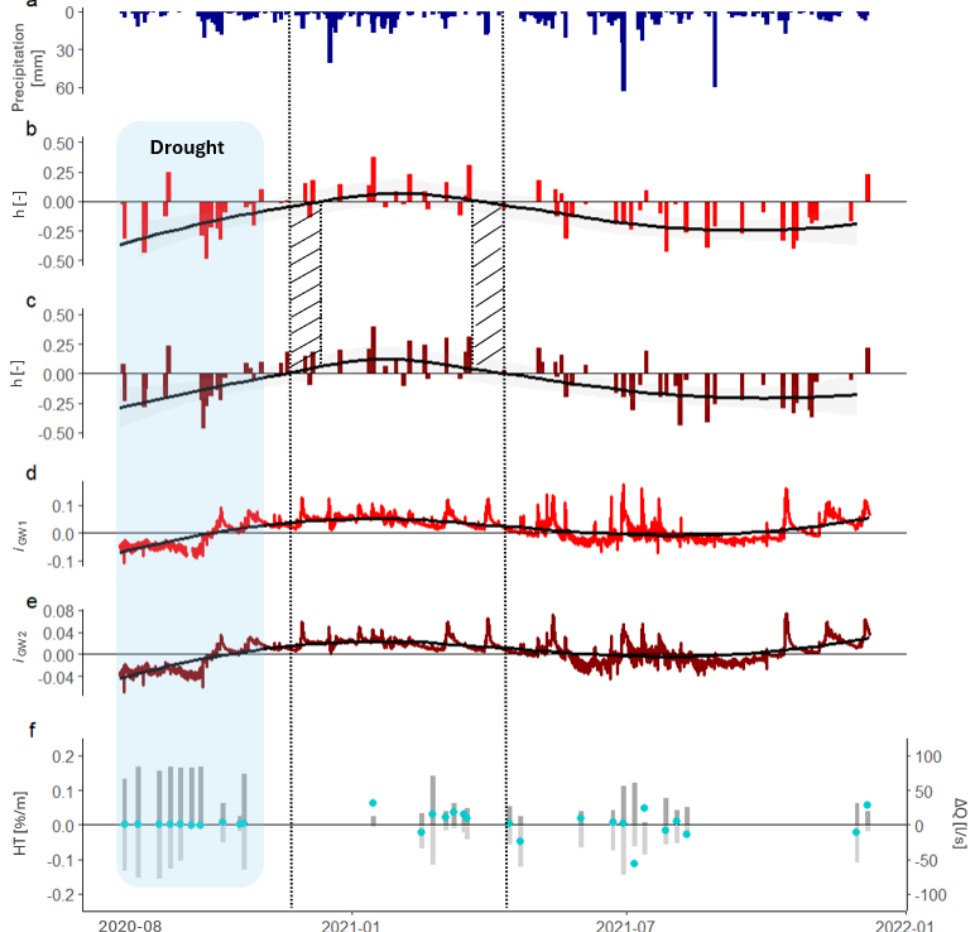

**Figure 4: (a) Overview of catchment precipitation (mm) derived from a near weather station. (b) h-Index of hysteresis between stream and GW1 (red), (c) h-Index of hysteresis between stream and GW2 (dark red) with moving average (black line) of index values (n=68) over the observation period. Hydraulic gradient between stream and GW1 (d), GW2 (e), with moving average (black line) of hydraulic gradients (Eq. 13) over the observation period. (f) Hydrological turnover (HT) in %/m point measurements (n= 28) during the observation period. Dotted black lines marking the transitions between summer and winter season. Blue background marks HT measurements during drought conditions (n=9).**





We observed extended HT during drought, and distinct seasonality of the h-Index (Figure 4) between winter and summer (including the drought period), with mostly positive index values in summer and negative values in winter. The shift between seasons of the h-Index corresponds with the crossing of its moving average of the zero line (Figure 4 b &c), with h-values

during the summer mostly negative (counterclockwise hysteresis) and during the winter positive (clockwise hysteresis) (Figure 2). However, there is a shift in seasonality between the two groundwater wells, with the well closer to the stream (GW1) showing the seasonal shift earlier and transitions later in the year compared to the second well (GW2). The distance between wells is 2.2m, which appears to correspond to one month in delay. Summer rainfall events show a marginal impact on low flows, just initializing sharp spikes in stream flow for a very short period (Figure 3). During winter, frequent rain

events sustain higher baseflow (Figure 3 b). However, the Olewigerbach is during both seasons a highly reactive stream with increased discharge amplitudes (Figure 3 b). Seasonality, reflected in precipitation and soil moisture data, aligns with h-index values and hydraulic gradients at the Olewigerbach (Figure 4).

### 3.1 Hysteresis Index in relation to HT

Comparing the h-indices, both wells at the experimental site show significant differences (Wilcoxon Signed-Rank Test) in h-

indices between summer and winter (Figure 5).

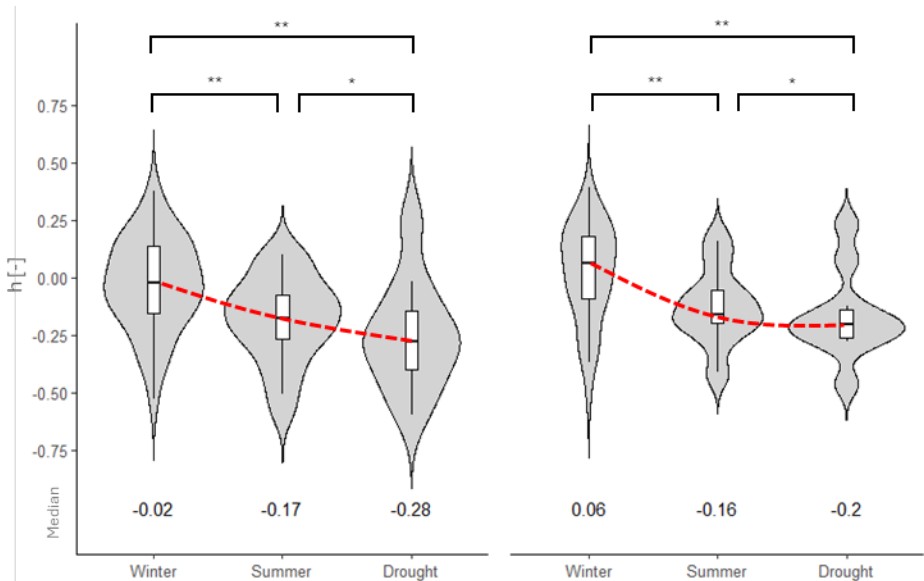

**Figure 5: (Left) Violine charts of h-index values (GW1), below median value of the integrated boxplot per season. (Right) Violine charts of h-index values (GW2), below median value of the integrated boxplot per season. Comparison between Winter (n= 37), summer (n=22), and drought (n= 9). Significant difference between groups indicated by * (*** p = 0.001, ** p =0.01, * p = 0.05, Wilcoxon Signed-Rank Test). Red dotted line showing decline in median h-index value between seasons.**



The first well tends towards more negative h-index values (counterclockwise) compared to the second well. Seasonally, both wells show significant differences between h-index values. During the summer period, the h-Index values present decreasing values, shifting from clockwise to counterclockwise behavior (Figure 5). Additionally, comparing the h-index

values tend further towards negative values during the drought periods. However, the drought dataset is small in observed events (n= 9). h-Index values are significantly elevated during drought compared to winter values. Thus, showing the increased extended duration of the total counterclockwise hysteresis between the stream water level and the groundwater level. Comparing summer HT to drought event HT, the significance level of the Pearson correlation decreases from summer to drought (Figure 5). Comparing the h-index values of both wells to HT, as the part of discharge exchanged per distance

(%/m), both wells behave similar. h-Index correlates significantly to HT (Figure 6a & b).

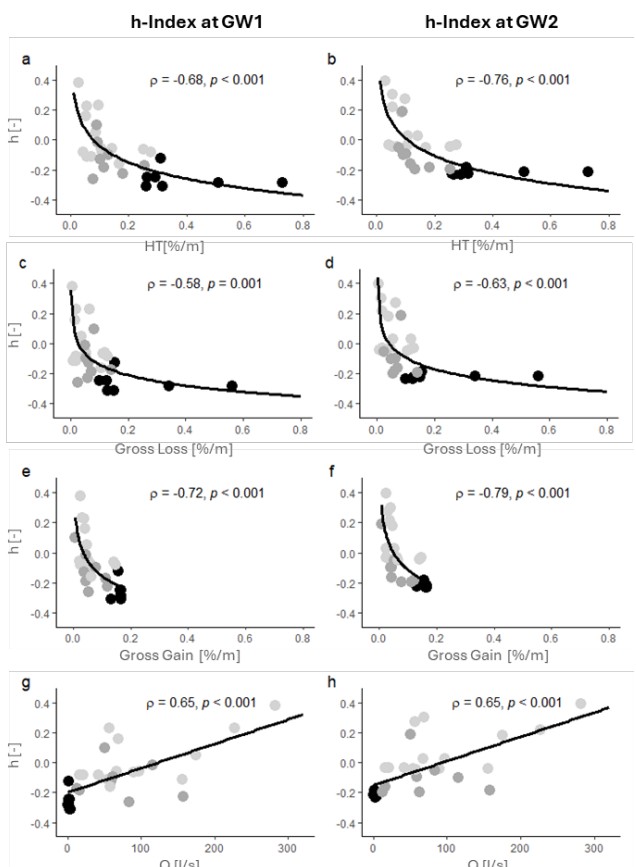

**Figure 6: Spearman correlation between h-index value (GW1: left, GW2: right) and HT (a, b), gross loss (c, d), gross gain (e, f) in %/m and discharge (g, h) in l/s. Black data points for drought (n= 7), dark grey for summer (n= 8) and light grey for winter (n= 14).**

Also, for the components of HT, gross loss (Figure 6c & d) and gross gain (Figure 6e & f). A comparable pattern is evident across all presented correlations: drought measurements cluster at the lowest levels of the h-index range, showing the highest observed HT values (Figure 6). These drought measurements were taken during periods of lowest discharge, whereas winter



measurements correspond to periods of highest discharge. Additionally, we observe a positive correlation between discharge
and h-index values at both wells (Figure 6g & h).

## 3.2 Hysteresis Index in relation to Hydraulic Gradients

The mean hydraulic gradient between groundwater wells and the stream during single events, defined as negative toward the

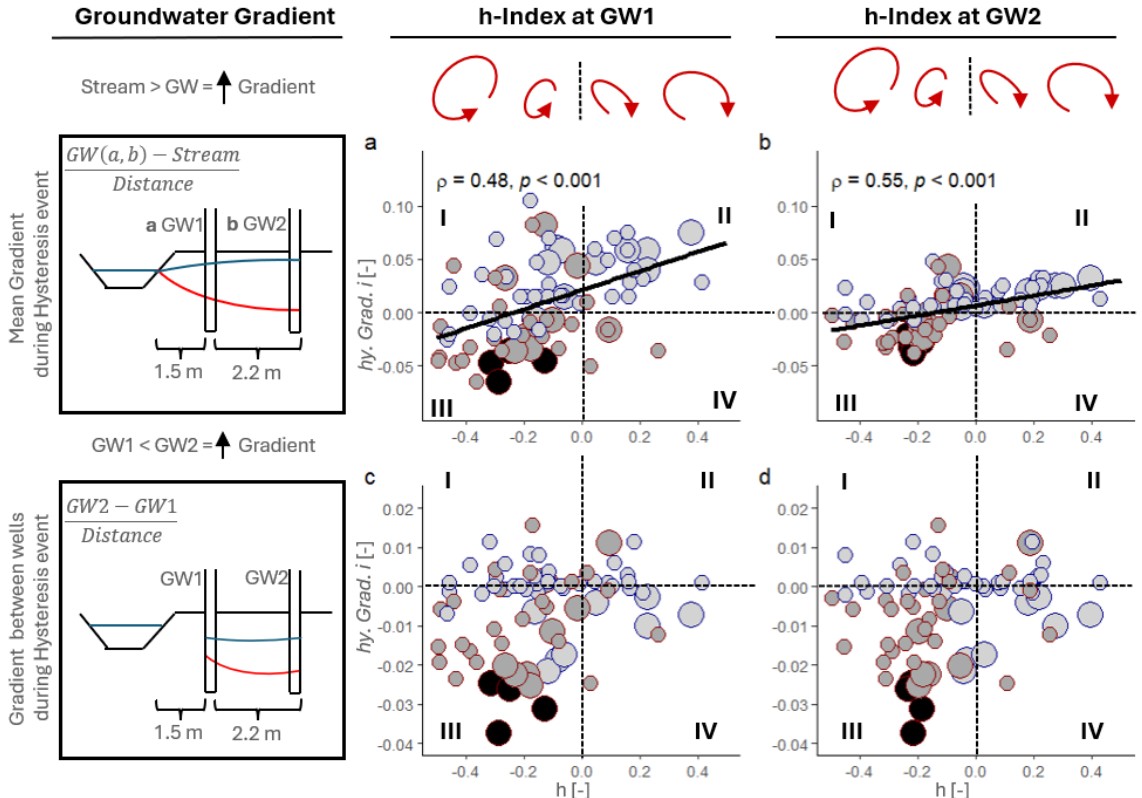

**Figure 7: Left: Schematic of hydraulic gradient. Right: Hysteresis direction indicated by red arrows. Mean hydraulic gradient against h-index values at GW1 (a) and GW2 (b) of the hysteresis event (n=68). Hydraulic gradient between GW1 and GW2 against h-index values at GW1 (c) and at GW2 (d). Dot size indicating corresponding HT measurements. Light gray blue couture (Winter), dark grey and red couture (Summer), black (drought). Plots are separated in quadrants: (I) Gaining & counterclockwise; (II) Gaining and clockwise; (III) Loosing & counterclockwise; (IV) Loosing & clockwise hysteresis.**

stream, shows a significant negative correlation with h-index values (Figure 7a & b; Eq. 12). Steeper negative gradients and more negative h-index values were primarily observed during summer and drought events. In contrast, winter events
exhibited both clockwise and counterclockwise hysteresis under less steep gradients. Positive gradients generally indicate gaining conditions, while negative gradients reflect losing conditions. Between the wells (Figure 7c & d; Eq. 13), most summer and drought events displayed negative gradients and counterclockwise hysteresis. Based on hydraulic gradient and h-index we deivided the observations into four behavioural clusters (I–IV):



- **Cluster I**: Mostly Winter events and heavy summer rain events; characterized by flat or positive gradients and counterclockwise hysteresis (GW rises before SW).
- **Cluster II**: Winter events; positive gradients with clockwise hysteresis (SW rises before GW).
- **Cluster III**: Predominantly summer and all drought events; negative gradients and counterclockwise hysteresis.
- **Cluster IV**: Smallest group; characterized by negative gradients and clockwise hysteresis; stream-GW pairs include four summer events with negative gradients and clockwise hysteresis. Between wells, five winter and two summer events shared these characteristics.

Gradients between wells were generally smaller and showed higher short-term variability during rain events, compared to stream-GW gradients.

### 3.3 Hysteresis Index and Stream Chemistry

Hysteretic behavior and the associatet HT induced mixing through the hyporheic zone and the stream bank, allow for

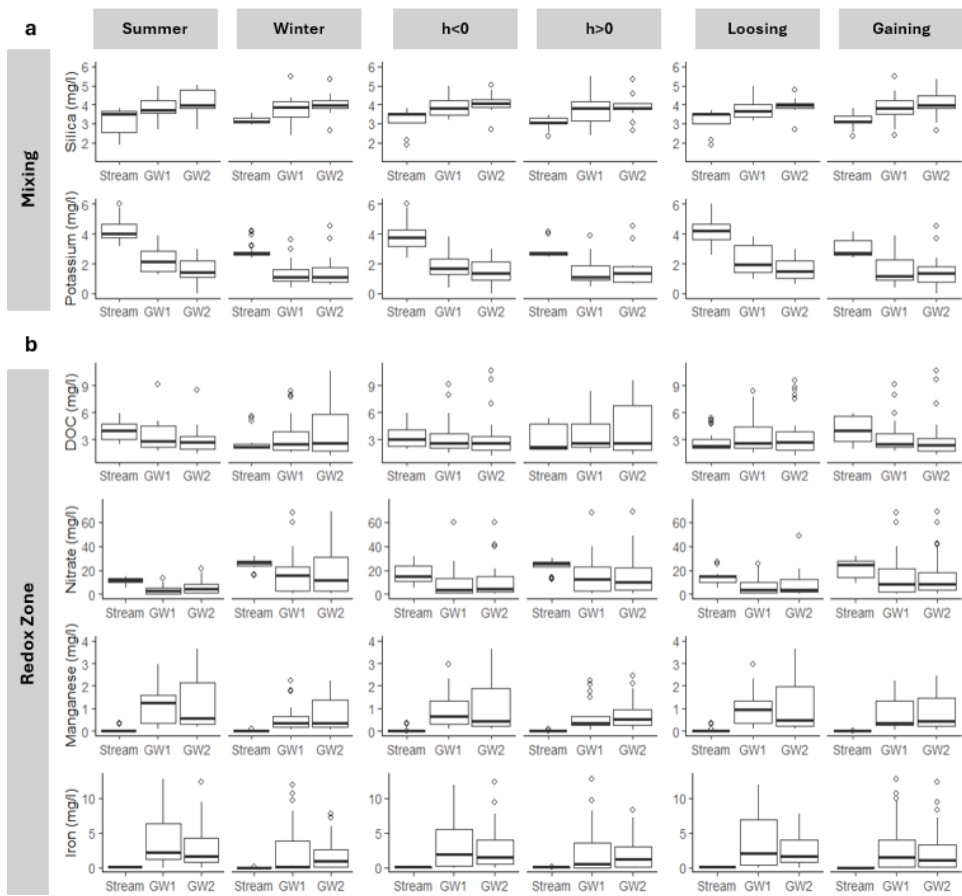

**Figure 8: Boxplot of water chemistry divided by season (Summer / Winter), hysteresis direction (h<0 / h>0) and hydraulic conditions (Gaining/Loosing). (a) Mixing proxies GW (Silica) and SW (Potassium). (b) Redox chain DOC, Nitrate, Manganese, Iron. Transect between Stream, GW1 and GW2, n= 48.**





exchange of solutes between GW and SW. Hence, we compared the sampling results by distinguishing seasons (Summer and Winter), Hysteresis direction (h<0, h>0) and hydraulic conditions based on the gradient data (Loosing or Gaining) (Figure 8). We observed differences in distribution of potassium samples between seasons. The winter events, positive h-Index show consistently the highest values in the stream with similar concentrations in both wells. However, for summer events, negative h-Indices and losing conditions exhibit high potassium values at the stream gradually declining with distance towards the

wells. In winter potassium is in a similar range at the wells (Figure 8a). Generally, silica is rising from groundwater to stream in concentration while potassium shows the opposite (Figure 8a). The redox sensitive species were observed in all samples in varying concentrations, with the highest concentrations in the near stream groundwater. For DOC concentrations tended to decline from stream to GW during the summer months while the opposite occurred during winter. Nitrate concentrations were generally variable with, on average, higher stream concentrations compared to GW (Figure 8b). Nitrate is constantly

present during our observations, regardless of redox potential indicated by the concentrations iron and manganese ions. Manganese and iron are also simultaneously present in comparably high concentrations at the GW wells and show distinct patterns. During summer and the corresponding system states GW1 exceeds the GW2 concentration median showing increased redox potential/activity directly in the riparian zone of the stream. During summer, losing conditions and with counterclockwise hysteresis manganese and iron concentrations are highest at GW1.

**4 Discussion**

We utilized the hysteresis index introduced by Zuecco et al. (2016) to analyze changes in the direction and magnitude of hysteresis between stream water levels and groundwater levels at two wells at the Olewigerbach catchment. High-resolution hydraulic gradient data allowed comparison of streamflow events with sufficient resolution to minimize noise. This allowed us to find seasonal variations in hysteretic patterns during the observation period comparable to those found in the literature

(e.g., Zuecco et al., 2016; Lloyd et al., 2016 a; Gelmini et al., 2022).

Although the h-index showed only slight differences in hysteresis between the wells (Figures 2–4), correlation analysis reveals significant spatial and temporal variability in groundwater-stream interactions (Figures 5 & 6). These variations reflect the dynamic nature of the hyporheic zone, which changes seasonally, particularly between winter, summer, and drought conditions (e.g., Wondzell, 2011; Cardenas, 2015; Harvey & Bencala, 1993; Malzone et al., 2016; Wroblicky et al.,

1998). As observed by Gelmini et al. (2022), our study identifies clear seasonality in hysteretic behavior. Specifically, hydrological turnover (HT) varies with seasonal changes in hydraulic gradients and discharge magnitudes (Figures 6 & 7). The h-index serves as an event-based parameter to describe GW-SW interactions by incorporating hydraulic preconditions and event characteristics. Distinct h-index value ranges correspond to increased portions of stream water engaged in turnover processes (Figure 6), aligning with studies emphasizing pressure-head variations and their influence on seasonal

groundwater-stream connectivity (Brunner et al., 2009, 2011).

The seasonal transition is characterized by declining h-index values (including directional changes) from winter to summer, correlating with reduced stream discharge and increased HT. This is consistent with findings from Payn et al. (2009), Covino





et al. (2011), and others. Beyond direction, the h-index also reflects temporal dynamics between groundwater and surface water levels during events (Zuecco et al., 2016; Lloyd et al., 2016 b), offering a physical framework for understanding HT

processes. Hydraulic gradients between the stream and shallow groundwater serve as preconditions for runoff events, modulating GW-SW interactions (e.g., Zimmer & McGlynn, 2017; Volz et al., 2013).

A distinct seasonal pattern emerges, steeper hydraulic gradients during summer events correspond to counterclockwise hysteresis, while gentler gradients in autumn and winter are associated with clockwise hysteresis (Figure 7). Soil moisture controls the unsaturated hydraulic conductivity, governing occurrence of infiltration and overland flow (Horton, 1933). Thus,

controlling threshold behavior and the hydraulic response to rainfall events (e.g. Zehe & Sivapalan, 2009). Based on these hypotheses we identified three system states during the observations, winter, summer and drought conditions. These patterns reflect groundwater storage states and hyporheic zone variability. Hydraulic gradient changes not only influence hysteretic loop direction but also reflect groundwater table dynamics, hyporheic zone extent, and riparian bank storage (e.g., Brunner et al., 2009; Gu et al., 2012; Malzone et al., 2016). Elevated stream stages promote counterclockwise hysteresis and hyporheic

zone expansion through increased bankstorage (Gu et al., 2012). Conversely, groundwater discharge initially reduces the hyporheic zone's extent (Cadenas & Wilson, 2007). Our results demonstrate that dynamic local gradients strongly influence hyporheic exchange fluxes and HT magnitude (Figure 7).

The observed clustering of hysteresis index and hydraulic gradients reveals distinct hydrological states driven by seasonal conditions and event timing. We show that plotting the events with hydraulic gradient against h-Index value four behavioral

clusters can be identified (Figure 7). Cluster I is dominated by winter events with counterclockwise hysteresis and flat or positive gradients, suggests storage influence where groundwater responds ahead of streamflow. This reflects a hyporheic reaction during early infiltration phases. Cluster II, with winter-dominated events and clockwise hysteresis under gaining conditions, likely indicates surface runoff or upstream input overwhelming the local GW response. These conditions reduce the influence of subsurface exchange, leading to stream-dominated hysteretic behavior. Cluster III shows the strongest

expression of hyporheic connectivity under summer and drought conditions. Steep negative gradients and counterclockwise loops here point to deeper, prolonged infiltration into the Stream bank enabling intense mixing, reflected by the high HT values during those conditions (Figure 6). Cluster IV shows the temporal variability and spatial heterogeneity of riparian processes. Particularly, the flatter gradients between the two wells are more susceptible to transient fluctuations during rainfall events, resulting in mixed hysteresis responses. However, this cluster fitted the least number of events during our

observed period (Figure 7 & 9). These findings show that combining h-index and hydraulic gradients allows detailed classification of hydrological states at event scale. Also reflected in HT as an independent measure of exchange. Counterclockwise hysteresis and high HT result in additional riparian bank storage. Vertical expansion of the hyporheic zone at Olewigerbach is likely limited by shallow sediments and small stream size (Malzone et al., 2016). Instead, horizontal bankstorage expansion dominates, forming storage capacities (Figure 9) along the streambank (Gu et al., 2012). The

occurrence of bank storage is associated counterclockwise hysteresis between stream water and groundwater levels, which occurs primarily during summer (Cluster III), while winter is dominated by clockwise hysteresis patterns (Figure 9). Thus,



the Olewigerbach exhibits distinct seasonal system states that shape the water volume at the surface-groundwater boundary layer and thus control the magnitude of HT as well.

We propose addressing HT as a proportion of total discharge, considering the extent of hyporheic zone involvement relative
to streamflow conditions. Under low surface discharge, hyporheic zone expansion potentially increases the ratio of hyporheic area to streamflow, engaging a greater fraction of stream water in HT. This is further promoted by the losing conditions observed at the site, as similarly described in the Selke catchment (Trauth et al. 2015) This promotes biogeochemical activity, as fluctuating water stages create alternating oxic and anoxic conditions, evident by the ions present during such conditions, especially at GW1 (Figure 8). Such conditions influence chemical turnover, pollutant degradation
and solute transport.

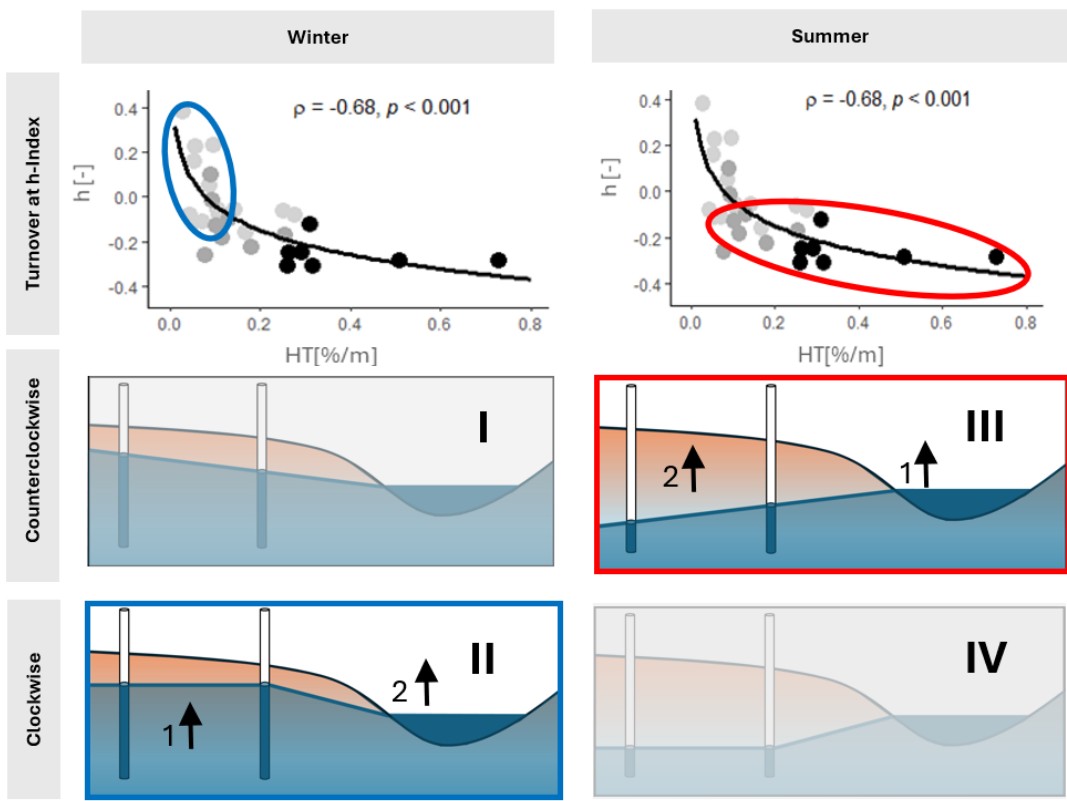

**Figure 9: Conceptual comparison of summer and winter conditions at the study site. Top: Seasonal Turnover at h-Index (Summer in red; Winter in Blue). Sketch of the Stream with adjacent groundwater Profile, under clockwise conditions and counterclockwise hysteric conditions. Red frame marking typical summer condition and blue frame marking typical winter**

In redox-sensitive environments, such as the riparian zone microbial reduction processes are understood to occur in a sequence with electron acceptors becoming progressively depleted (Zehnder & Stumm, 1988). This sequence is influenced by factors such as redox potential, pH, electron acceptor availability, and the presence of bioavailable organic matter.



During summer and under counterclockwise hysteresis conditions, we observed increased microbial activity and oxygen depletion within the riparian zone, likely driven by enhanced inflow of labile organic carbon from the stream into near-stream sediments (Smith & Arah, 1986). This is evidenced by elevated DOC concentrations and redox-sensitive solutes at GW1 (Figure 8). However, our observations revealed the co-occurrence of nitrate, manganese, and iron in groundwater and stream samples. This simultaneous presence of multiple electron acceptors suggests that the redox sequence is not strictly adhered to under field conditions a finding consistent with Alewell et al. (2008) and others. Such deviations from idealized zonation likely result from heterogeneous flow paths, dynamic hydraulic gradients, and continuous mixing at the GW-SW interface (Gu et al., 2012). Such conditions foster overlapping redox states, enabling parallel rather than sequential redox processes. This highlights the riparian zone as a hydrologically and biogeochemically active layer, supporting diverse redox environments (e.g. Hill & Cardaci, 2004; Carlyle & Hill, 2001). Our findings emphasize the coexistence of multiple redox processes. The resulting shifts between oxic and anoxic conditions are critical for degrading pollutants and supporting biogeochemical activity in the riparian zone. These dynamics underscore the importance of water-stage fluctuations in enhancing chemical exchanges and microbial activity at the GW-SW boundary (Gu et al., 2012).

## 5 Conclusion

Seasonal variability in GW-SW interactions at the Olewigerbach is driven by changes in discharge, hydraulic gradients, and hyporheic zone dynamics. Our study highlights the h-index as a diagnostic tool for quantifying seasonal shifts in GW-SW interactions, particularly its strong correlation with HT, shedding light on the mechanisms shaping hyporheic zone dynamics. We characterized the hysteresis between stream water levels and groundwater levels in two riparian wells over two years, calculating the h-index for 68 events and conducting HT estimations for 28 of these events. The h-index revealed clear distinctions between summer and winter conditions, with drought events emerging as statistical outliers. Importantly, we found a strong correlation between hysteretic behavior, quantified through the h-index, and HT. This highlights the role of the h-index in linking hydrological processes with turnover dynamics, particularly during low-flow conditions. Seasonal hydraulic gradients and their influence on bank storage created additional mixing and reaction spaces, increasing the stream water contribution to the hyporheic zone. Elevated HT aligns with hysteresis loop direction and extent, highlighting the hyporheic zone's dynamic spatial reorganization and its influence on solute transport penetrating the riparian zone and creating microenvironments with overlapping redox states.

Our findings demonstrate that hysteresis behavior marks HT-driven changes in hyporheic zone volume and significantly increase streamflow engagement with the underground, especially during summer droughts. This expansion facilitates enhanced mixing processes, influencing stream chemistry. The chemical transects from the stream through the groundwater wells reinforced the critical role of hydraulic gradients in modulating mixing and connectivity within the riparian zone. Thus, leading to frequent shifts between oxic and anoxic conditions, critical for degrading pollutants and supporting biogeochemical activity.



In the context of future climate change, where prolonged droughts and reduced discharge magnitudes are likely to become more frequent, our study suggests that the hyporheic zone will increasingly sustain nutrient cycling and be a prominent component of GW-SW connectivity in gravel bed rivers throughout central Europe. The patterns of elevated HT and large gross exchanges observed during drought conditions indicate that turnover-induced mixing may dominate solute exchange and reaction processes under such scenarios. Our assessment illustrates a robust framework for advancing our understanding of solute transport and GW-SW dynamics at lower order streams, aiding water management under changing climatic conditions.

**Author contributions**

LB and TS conceptualized the study. Further, LB collected and analysed field and Lab data. Both LB and TS contributed to the final version of the manuscript.

**Competing interests**

The authors declare that they have no conflict of interest.

**Acknowledgments**

The Authors thank Sven Ulrich for his help during initial data acquisition. The author Lars Bäthke is funded by a PhD-stipend of the Studienwerk Villigst e. V., Germany.

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
