# Peer review of "Hysteresis between groundwater and surface water levels indicates the states of hydrological turnover affecting solute transport and redox processes"

_EGUsphere, 2025_

## Author Comment (AC4)

**Response to Reviewer 2**

We would like to thank Reviewer 2 for the constructive and detailed comments, which helped us to further improve the clarity and robustness of our manuscript.

**General comment**

*"...very limited number of groundwater wells, which may not be representative... such limitation should be discussed..."*

**Response:**

We acknowledge the reviewer's concern regarding the limited number of groundwater wells and agree that this represents a potential limitation. However, at the studied catchment there are no groundwater wells other than the near stream monitoring wells utilized in this study. Geological conditions of that catchment do not provide a continuous, near surface groundwater aquifer. Hence, in this catchment with fissured schist bedrock and comparably shallow soils, alluvial groundwater wells are most likely suitable to capture stream–groundwater connectivity. Hence, we relate to the soil moisture model as an alternative proxy for catchment saturation conditions, since larger scale groundwater monitoring is absent.

Our analyses are based on one stream section and two groundwater wells, which we will now explicitly acknowledge in the discussion. We clarify that the findings reflect local hydrogeological conditions and should not be considered directly transferable to other parts of the catchment. At the same time, we emphasize that the conceptual framework developed (h-index/HT relationships) is intended as a process-based example illustrating hydrological connectivity, rather than a catchment-wide quantification.

We will add the following sentence to the discussion chapter of the manuscript:

> *"This study is relying on two available groundwater wells located in the close vicinity of the stream reach under study. While these wells provide valuable process-based insights into GW–SW exchange and redox dynamics, they cannot capture spatial heterogeneity across the catchment. The geological setting with fissured schists as the near surface bedrock does not provide a contiguous shallow aquifer, and thus no groundwater monitoring network at the catchment-scale. Therefore, larger-scale GW monitoring is absent. We instead utilize the standardized soil moisture model provided by the public weather service for each weather station (DWD) as an alternative proxy for subsurface storage dynamics. Consequently, the conceptual framework we propose (h-index/HT relationships) should be understood as locally constrained and hypothesis-generating, rather than directly representative of larger scales."*

*"...a characterization of the runoff events is currently missing..."*

**Response:**
We acknowledge the reviewer's suggestion. Precipitation characteristics and corresponding streamflow responses are already included in Figures 3 and 4, where rainfall is shown as bar plots alongside discharge. We will clarify this in the caption and Methods section to make it easier for the reader to connect runoff dynamics with our event analysis. While descriptive statistics (rainfall

amounts, intensities, duration, peak discharge, antecedent wetness) are useful context, our classification of events is based on hysteresis behaviour and hydrological turnover, rather than on event magnitude or precipitation metrics. We will therefore add a short descriptive summary of event characteristics in the Methods text but maintain the focus of our framework on hysteresis-based classification.

> *"During the observation period (2021–2023), we monitored 68 runoff events with precipitation totals ranging up to 60 mm. Corresponding streamflow responses showed peak discharges between 20 and 700 L s $^{-1}$. Antecedent soil moisture conditions varied from dry at summer drought events to near saturation at winter events (Fig.3). While these event descriptions provide hydrological context, the classification of events in this study is based on hysteresis behaviour and hydrological turnover rather than on precipitation or discharge metrics."*

*"...specific objectives should be clarified at the end of the introduction, and results/discussion should be organized based on these objectives."*

**Response:**
As suggested by Commentator 1 and Reviewer 1, we now explicitly state our objectives and hypotheses at the end of the Introduction.

> *"The objective of this study is to explore how seasonal changes in hydraulic conditions influence groundwater–surface water interactions in a headwater stream. We hypothesize that (1) the direction and magnitude of hysteresis between groundwater and stream water levels are indicative of seasonal hydrological states, and (2) consequently, the direction and magnitude of the hysteresis is related to hydrological turnover. thus facilitating (3) redox fluctuations creating oxic-anoxic transition in the riparian zone, shaping redox hot spots."*

We will revise our Results and Discussion sections to be restructured accordingly, to better align with these objectives and hypotheses.

**Specific comments**

- *Line 12: Please explain which variables are used for hysteresis analysis.*

  **Response:** Clarified in Methods: hysteresis analysis was performed using stream stage and groundwater levels (GW1, GW2), normalized to facilitate comparison across events.

  *We will clarify the sentence in line 12 accordingly:*

  > *"This study examines the relationship between hydrological turnover (HT) and stream-stage-groundwater-level hysteresis patterns under various system states in a third-order tributary of the River Mosel in Trier, Germany, using high-resolution stream-stage and groundwater-level data (GW1, GW2) together with complementary chemical observations collected over two years."*

- *Introduction: The specific objectives of this study are not clearly presented. Furthermore, the authors should consider reporting their research hypotheses, which are later mentioned at Line 306 (Discussion).*

    **Response:** Addressed as above; hypotheses are now clearly stated at the end of the Introduction (see also responses to Reviewers 1 & Commentator 1).

- *Materials and methods: details about the selected events... rainfall amounts, intensities, etc.*

    **Response:** We agree that event details are beneficial to the study. We will add a new table presenting the suggested details for rainfall events and streamflow response prior to HT estimation experiments. After the presented events, one or multiple HT experiments were conducted. The complete list of all monitored events will be provided as supplementary material.

    *Characteristics of events occurring prior to corresponding tracer experiment*

| Date | Duration [h] | Precipitation [mm] | Mean Q [L/s] | Runoff vol. [m³] | Runoff coeff. [–] |
|---|---|---|---|---|---|
| 03-08-2020 | 70 | 3.8 | 13.5 | 3387 | 0.111 |
| 17-08-2020 | 24 | 8.3 | 14.5 | 1259 | 0.019 |
| 18-08-2020 | 48 | 2.2 | 0.1 | 10 | 0.001 |
| 02-09-2020 | 26 | 3.0 | 13 | 1240 | 0.596 |
| 24-09-2020 | 33 | 8.0 | 29.9 | 3529 | 0.056 |
| 06-10-2020 | 24 | 16.3 | 47.6 | 4110 | 0.031 |
| 09-10-2020 | 289 | 22.6 | 25.6 | 26684 | 0.148 |
| 23-10-2020 | 50 | 8.2 | 13.9 | 2524 | 0.039 |
| 08-02-2021 | 84 | 19.5 | 165.5 | 49981 | 0.321 |
| 13-03-2021 | 29 | 18.2 | 61.2 | 6336 | 0.044 |
| 19-03-2021 | 337 | 30.9 | 42.8 | 51882 | 0.210 |
| 11-04-2021 | 588 | 42.1 | 36.5 | 77294 | 0.229 |
| 26-05-2021 | 116 | 15.3 | 14.8 | 6171 | 0.050 |
| 04-06-2021 | 71 | 19.8 | 18.6 | 4777 | 0.030 |
| 29-06-2021 | 122 | 7.0 | 4.9 | 2159 | 0.039 |
| 05-07-2021 | 51 | 18.1 | 38.7 | 7114 | 0.049 |
| 27-07-2021 | 150 | 28.2 | 80.1 | 43289 | 0.192 |
| 08-08-2021 | 31 | 4.2 | 16.7 | 1861 | 0.056 |
| 26-11-2021 | 85 | 15.4 | 17.6 | 5376 | 0.044 |
| 07-12-2021 | 557 | 29.4 | 95.3 | 190892 | 0.813 |

- *Lines 114–123: belong elsewhere.*

  **Response:** We will move this text to the Experimental setup subsection.

- *Lines 139–141: provide NaCl injection details.*

  **Response:** We will add sentences and provide additional information about the monitored stream section, the multiple injection points, distances from wells, and timing.

  *We will revise the sentence as follows:*

  > *"Instantaneous NaCl tracer injections were performed for HT estimation, during recession, utilizing salt dilution gauging (Day, 1976; Covino et al., 2011; Mallard et al., 2014). At the monitored stream section (~250 m) two independent discharge measurements were conducted, at the upstream and downstream ends of the stream reach. Injection distance for the measurement devices was approximately 30m (Bäthke & Schuetz, 2024). The $Q_{loss}$ estimation is based on an additional tracer injection injected at the upstream site and measured at the downstream site, following the instructions of Payn et al. (2009). The wells are located in 1.5 m and 3 m lateral distance from the upstream measurement point. Electrical conductivity during tracer injections..."*

- *Lines 155–156: When were samples taken?*

  **Response:** We will add sentences and provide additional information about the timing of water sampling.
  We will revise the sentence as follows:

  > *"Groundwater and stream samples were collected during stream flow recession towards baseflow conditions, immediately prior to the dilution gauging experiments (n = 48, table xx) and not during rainfall events. Laboratory analyses were carried out after filtration through a 0.45 μm glass fiber filter (Macherey & Nagel). Nitrate was determined by ion chromatography (±2%), potassium (±5%) and silica (±8%) by ion chromatography, iron and manganese (±5%) by atomic absorption spectroscopy (AAS, contrAA 300), and dissolved organic carbon (DOC) by combustion followed by IR detection (TOC analyzer, ±2%)."*

- *Lines 157–159: What do ± values indicate?*
  **Response:** We will add clarification about the meaning of the ± values.

  We will revise the sentence as follows:

  > *"The reported ± values represent the standard deviation of replicate analyses."*

- *Equation 3: How is $Q_{gain}$ determined?*

   **Response:** $Q_{gain}$ is calculated by combining discharge and conservative solute mass balances along the reach. Starting with determination of Q at the start and the end of the stream section. Upstream and downstream discharge measurements are used to determine the net discharge change (ΔQ), while the separate tracer measurement along the total stream- section allows for the determination of gross losses to the subsurface. The tracer-marked stream water losses are replaced by unmarked groundwater inflow. Balancing ΔQ and $Q_{loss}$ allows $Q_{gain}$ calculation We will add an explanatory sentence to the Methods section accordingly.

   > *"The absolute differences in Q (ΔQ, eq. 3) between upstream and downstream discharge measurement sites in combination with $Q_{Loss}$ (eq. 2) results in $Q_{gain}$ as the residual, i.e. the part of ΔQ which is not explained by $Q_{loss}$ (eq. 2)."*

- *Lines 191–195: A scheme with hysteretic loops would be helpful.*

   **Response:** As also suggested by Reviewer 1, we aim to add a schematic, illustrating clockwise and counterclockwise hysteresis loops and their interpretation.

- *Results should be structured according to objectives.*

   **Response:** Revised structure: Results now will follow the three stated objectives (Hysteresis behaviour → Hydrological turnover → Solute/redox dynamics) according to suggestions of Commentator 1 and Reviewer 1.

- *Section 3.3: Expand link between hysteresis and solutes.*

   **Response:** We will expand this section leading to the discussion on tracer behaviour (Reviewer 1), event-specific differences, and rationale for grouping tracers into mixing vs. redox-sensitive groups (linking to revised hypotheses).

   > *"We grouped the tracers into a conservative mixing set (geogenic silica, Potassium) and a redox-related set to reflect the hypotheses 1&2 as well as 3, respectively: mixing tracers diagnose hydrological turnover and its seasonal variability in groundwater-surface water exchange. This separation allows redox tracers (DOC, $NO_3^-$, Fe, Mn) to specifically indicate the biogeochemical response to that exchange. Redox tracers can track redox dynamics as conceptualized by biogeochemical hydrological coupling (Peiffer et al., 2021) and the classical redox sequence in oxygen-limited environments (Zehnder & Stumm, 1988). We emphasize that event metrics are used as hydrological context, while event classification itself is based on hysteresis behaviour."*

- *Figure 8: Are differences significant?*

  **Response:** We will add significance indicators within the Graphics of Figure 8.

- *Line 306: Hypotheses should be reported earlier.*

  **Response:** We agree and will state the hypothesis in the introduction.

- *Discussion: lacks structure, does not address limitations.*

  **Response:** We will revise the Discussion to improve clarity and structure. The revised Discussion will explicitly address both limitations (e.g., limited number of wells and hydrological turnover events) and broader implications. It will be organized into the following sections: (i) hysteresis direction and magnitude as indicators of seasonal hydraulic states, (ii) hydrological turnover, (iii) solute and redox dynamics.
* * *
We thank the reviewer for reminding us of such technical oversights. We will change the manuscript accordingly by incorporating all the suggestions below:

- Lines 77–78: Rephrased for clarity.
- Line 105: "mouth" replaced by "outlet".
- Figure 3 caption: explained "mQ".
- Figure 5: number of samples added; caption revised to "violin plots".
- Line 241: comma inserted.
- Line 253: corrected "deivided" → "divided".
- Figure 8: added number of samples per boxplot.
- Line 267: inserted missing "for".
- Line 330: inserted missing "with".